# Sickle Cell Disease—Genetics, Pathophysiology, Clinical Presentation and Treatment

**DOI:** 10.3390/ijns5020020

**Published:** 2019-05-07

**Authors:** Baba P. D. Inusa, Lewis L. Hsu, Neeraj Kohli, Anissa Patel, Kilali Ominu-Evbota, Kofi A. Anie, Wale Atoyebi

**Affiliations:** 1Paediatric Haematology, Evelina London Children’s Hospital, Guy’s and St Thomas NHS Trust, London SE1 7EH, UK; 2Pediatric Hematology-Oncology, University of Illinois at Chicago, Chicago, IL 60612, USA; 3Haematology, Guy’s and St Thomas NHS Trust, London SE1 7EH, UK; 4Holborn Medical Centre, GP, London WC1N 3NA, UK; 5Paediatrics Department, Basildon and Thurrock University Hospitals, NHS Foundation Trust, Basildon SS16 5NL, UK; 6Haematology and Sickle Cell Centre, London North West University Healthcare NHS Trust, London NW10 7NS, UK; 7Department of Clinical Haematology, Cancer and Haematology Centre, Oxford University Hospitals NHS Foundation Trust, Churchill Hospital, Oxford OX3 9DU, UK

**Keywords:** sickle cell disease (SCD), pathophysiology, hydroxyurea/hydroxycarbamide, haemolysis, vaso-occlusive crisis, acute chest syndrome, end-organ damage, bone marrow transplant, anaemia, foetal haemoglobin, gene therapy for haemoglobinopathies

## Abstract

Sickle cell disease (SCD) is a monogenetic disorder due to a single base-pair point mutation in the β-globin gene resulting in the substitution of the amino acid valine for glutamic acid in the β-globin chain. Phenotypic variation in the clinical presentation and disease outcome is a characteristic feature of the disorder. Understanding the pathogenesis and pathophysiology of the disorder is central to the choice of therapeutic development and intervention. In this special edition for newborn screening for haemoglobin disorders, it is pertinent to describe the genetic, pathologic and clinical presentation of sickle cell disease as a prelude to the justification for screening. Through a systematic review of the literature using search terms relating to SCD up till 2019, we identified relevant descriptive publications for inclusion. The scope of this review is mainly an overview of the clinical features of pain, the cardinal symptom in SCD, which present following the drop in foetal haemoglobin as young as five to six months after birth. The relative impact of haemolysis and small-vessel occlusive pathology remains controversial, a combination of features probably contribute to the different pathologies. We also provide an overview of emerging therapies in SCD.

## 1. Introduction

Sickle cell disease (SCD) was first reported by Herrick in 1910 even though reports suggest prior description of the disorder [1]; it is the result of homozygous and compound heterozygote inheritance of a mutation in the *β-globin* gene. A single base-pair point mutation (GAG to GTG) results in the substitution of the amino acid glutamic acid (hydrophilic) to Valine (hydrophobic) in the 6th position of the β-chain of haemoglobin referred to as haemoglobin S (HbS) [2]. Phenotypic variation in clinical presentation is a unique feature of SCD despite a well-defined Mendelian inheritance, the first to be molecularly characterised as described by Pauling [3] and confirmed to be due to a single amino acid substitution by Ingram [3] almost 70 years ago. SCD is a multi-organ, multi-system disorder with both acute and chronic complications presenting when foetal haemoglobin (HbF) drops towards the adult level by five to six months of age [4].

## 2. Classification

The inheritance of homozygous HbS otherwise referred to as sickle cell anaemia (SCA) is the most predominant form of SCD, the proportion varies according the country of origin [5,6,7]. The next most common form of SCD is the co-inheritance of HbS and HbC—referred to as HbSC, this is most prevalent in Western Africa, particularly Burkina Fasso and Mali and the coastal countries including Ghana, Benin and Western Nigeria [5,7,8]. The co-inheritance with β thalassaemia results in a sickle β thalassaemia genotype (*HbS/βo or HbS/β+*), depending on the genetic lesion on the thalassaemia component, the clinical presentation may be mild or equally as severe as homozygous SCD (HbS/HbS) [9]. Those with *HbS/βo*-thalassaemia have a more severe course of disease similar to homozygous SS patients, while offspring with *HbS/β+*-thalassaemia depending on the β-globin mutation is associated with variable phenotype from mild to severe phenotypes SCD [3,10].

## 3. Epidemiology

SCD is one of the most common inherited life-threatening disorders in human, it predominantly affect people of African, Indiana and Arab ancestry [5,11,12]. It is estimated that over 80% of over 300,000 annual births occur in sub-Saharan Africa (SSA), the largest burden from Nigeria and Democratic Republic of Congo [13]. The gene frequency is highest in West African countries with 1 in 4 to 3 (25–30%) being carriers of HbS compared to 1/400 African Americans and is variable in European populations [14,15,16]. The prevalence of SCD in developed countries is increasing partly due to migration from high prevalent countries [17,18,19,20]. It is estimated that over 14,000 people live with SCD in the UK, similar to France, while countries like Italy, Germany have seen increasing numbers from Africa [21,22,23,24]. With increasing survival, the age distribution of SCD is changing from a childhood disorder pattern that patients now survive into adulthood and old age. It is now reported that over 94% of those born with SCD now survive into adulthood in the US, France and UK in contrast to the high mortality in SSA where 50–90% may die in the first five years of life [12,25,26]. In low resource settings and countries where newborn screening is not yet standard care, patients may die young even before diagnosis is confirmed [27]. Among the common causes of death in the absence of early diagnosis followed by education and preventive therapies such as penicillin prophylaxis and regular surveillance include infections, severe anaemia (acute splenic sequestration, aplastic anaemia) and multi-organ failure [28]. It is essential therefore that Newborn and Early Infant diagnosis is given the priority it deserves by those countries where SCD is a public health problem [28,29].The implementation of early infant diagnosis remains out of reach for the majority of countries in SSA despite multiple declarations by international organisations and public statements by politicians to honour such commitments. The benefits for screening can only become meaningful when such practice is embraced by policy-makers across the continent and India where the majority of SCD are born and live. Comprehensive care includes penicillin V prophylaxis, Hydroxycarbamide therapy and preventive therapies such as antimalarials and health promotion where relevant will improve outcomes and health related quality of life [30].

## 4. Pathophysiology

The schematic representation in Figure 1 highlights the pathophysiology of SCD [31]. Red blood cells (RBCs) that contain HbS or HbS in combination with other abnormal β alleles, when exposed to deoxygenated environment undergo polymerisation and become rigid. The rigid RBC’s are liable to haemolysis, and due to increased density may affect blood flow and endothelial vessel wall integrity. The dense rigid RBC’s lead to vaso-occlusion, tissue ischaemia, infarction as well as haemolysis [32]. The consequence of haemolysis is a complex cascade of events including nitric oxide consumption; haemolysis linked nitric oxide dysregulation and endothelial dysfunction which underlie complications such as leg ulceration, stroke, pulmonary hypertension and priapism [33]. Unlike normal RBC’s with half-life of approximately 120 days, sickle RBC’s (sRBC) may survive just 10–20 days due to increased haemolysis [34]. During deoxygenation; healthy haemoglobin rearranges itself into a different conformation, enabling binding with carbon dioxide molecules which reverts to normal when released [32]. In contrast, HbS tends to polymerise into rigid insoluble strands and tactoids, which are gel-like substances containing Hb crystals. During acute sickling, intravascular haemolysis results in free haemoglobin in the serum, while RBC’s gaining Na^+^, Ca^2+^ with corresponding loss of K^+^ [31]. The increase in the concentration of Ca^2+^ leads to dysfunction in the calcium pump. The calcium depends on ATPase but it is unclear what role calcium plays in membrane rigidity attributed to cytoskeletal membrane interactions. [35]. Furthermore, hypoxia also inhibits the production of nitric oxide, thereby causing the adhesion of sickle cells to the vascular endothelium [33]. The lysis of erythrocytes leads to increase in extracellular haemoglobin, thus increasing affinity and binding to available nitric oxide or precursors of nitric oxide; thereby reducing its levels and further contributing to vasoconstriction [32].

## 5. Disease Modifiers

The sickle β-globin mutation renders the sickle gene pleiotropic in nature, with variable phenotypic expression associated with complex genetic and environmental interactions, as well as disease modifiers that are increasingly being recognised. Sickle RBC (sRBC) polymerisation in deoxygenated environments is influenced by a number of factors including the co-inheritance of alpha thalassaemia, foetal haemoglobin (HbF) level which is determined by a number of genetic factors including genetic variations of BLC11A, HBS1L-MYB and HBB loci and hydroxycarbamide therapy amongst others [35]. The BCL11A and ZBTB7A genes (LRF protein) are responsible for the suppression of γ chains, and HbF production. HbF reduces sickle cell polymerisation due to reduction in HbS concentration and the fact that it is excluded from the sickle cell polymers. HbF also has high oxygen retention thereby ameliorating both the vaso-occlusive and haemolytic pathology in SCD. Among the four main sickle haplotypes, the level of HbF is highest in the Indian/Arab haplotype [8] predominantly found in the Arab Peninsula and India followed by the Senegal, and Benin haplotypes most predominant in Sub-Saharan Africa. The Bantu haplotype with the lowest HbF is predominantly found in Central African countries [8,36]. Alpha thalassaemia also modulates the expression of SCD. People with one or two α genes deleted have less haemolysis and fewer vasculopathy complications [8].

## 6. Clinical Manifestations

SCD is characterised by protean manifestations ranging from acute generalised pain to early onset stroke, leg ulcers and the risk of premature deaths from multi-organ failure [8]. As a result of the effect of HbF, clinical features do not begin until the middle to second part of the first year of post-natal life when this has predominantly switched to adult haemoglobin [36,37,38,39,40].

### 6.1. Vaso-Occlusive Crisis (Pain)

Patients with SCD may experience intense pain early in infancy, childhood and adulthood. Pain usually accounts for the majority of hospitalisations and overall negative impact in patients’ health related quality of life. Pain is the cardinal feature of SCD and it is characteristically unpredictable, episodic in nature, described as one of the most excruciating forms of pain that affects human beings. Pain occurs due to stimulation of nociceptive nerve fibres caused by microvascular occlusion. The microcirculation is obstructed by sRBCs, thereby restricting the flow of blood to the organ and this results in (i) ischaemia, (ii) oedema, (iii) pain, (iv) necrosis, and (v) organ damage [10]. In the first year of life one of the cardinal features is the ’hand-foot syndrome’ due to vaso-occlusion of post-capillary vasculature resulting in tissue oedema and pain of the extremities [41]. Infants display their pain nonverbally with irritability and apparent ‘regression’ tendencies such as inability to weight bear, walk or crawl. In older children and adults, vaso-occlusive pain can affect any part of the body. The onset of pain is spontaneous, usually no precipitating factors; well-known triggers include infections, fever, dehydration, acidosis, sudden change in weather including wind speed, cold, rain and air pollution. Resolution of pain is unpredictable. Acute pain might lead to chronic pain [42].

### 6.2. Anaemia

Symptomatic anaemia is the commonest symptom in SCD generally more common in SCA (Homozygous S), which usually runs the lowest haemoglobin level common to double heterozygous states. The steady state haemoglobin for asymptomatic patients varies according to the phenotype, ranging from levels as low as 60–80 g/L for homozygous S and /*Sβo* to 100–110 g/L in double heterozygous *SC* and *Sβ*+ forms. However, the rate of fall from individual steady state haemoglobin level may trigger symptoms of hypoxia (aplastic crises) or a shock-like state (e.g., acute splenic sequestration) [4,43].

### 6.3. Acute Aplastic Crisis

The most common cause of acquired bone marrow failure in SCD and other haemolytic disorders is caused by Parvovirus B19 [44]. This virus causes the fifth disease and normally in healthy children is quite mild, associated with malaise, fever and sometimes a mild rash; the virus affects erythropoiesis by invading progenitors of RBCs in the bone marrow and destroying them, thus preventing new RBCs from being made. There’s a slight drop in haematocrit in children with HbAA who are generally unaffected, however in SCD as the lifespan of RBC’s is reduced to about 10–20 days, there is a significant drop in haemoglobin concentration. Parvovirus B19 infection usually takes about four days to one week to resolve and patients with SCD usually require a blood transfusion [39,44].

### 6.4. Infection

SCD increases susceptibility to infections, notably bacterial sepsis and malaria in children under five years [45]. Respiratory infections can trigger the sickle-cell acute chest syndrome, with a high risk of death. Risk factors for infections include: (i) functional asplenia/hyposplenia which present with reduced splenic immune response at a very young age, (ii) impaired fixation of complement, (iii) reduced oxidative burst capacity of chronically activated neutrophils, dysfunctional IgM and IgG antibody responses and defective opsonisation. The main pathogen of concern is *Streptococcus pneumoniae,* though severe and systemic infections arise with *Haemophilus influenzae*, *Neisseria meningitides*, and *Salmonellae* leads to osteomyelitis especially *Salmonella* due to bowel ischaemia and gut flora dissemination [46,47].

### 6.5. Splenic Sequestration Crisis

The main function of the spleen is the removal of defective red blood cells including sickled RBCs (sRBC) resulting in further haemolysis [48]. Blood flow through the spleen is slow reducing oxygen tension and increased polymerisation in HbS. As a result of the narrow capillaries in the splenic vascular bed, further hypoxia occurs with RBC polymerisation and entrapment of affected blood cells. This leads to a cycle of hypoxia, RBC polymerisation, and reduced blood flow causing the spleen to enlarge, for unexplained reasons, this may occur suddenly with pooling of blood within the vascular bed resulting in shock and circulatory failure. The rapidly increasing spleen size may lead to abdominal distension, sudden weakness, increased thirst, tachycardia and tachypnoea. Splenic sequestration crisis is an emergency because if left untreated, it can lead to death in 1–2 h due to circulatory failure [49].

### 6.6. Other Complications

The complications listed above are highlighted as those affecting babies and young children, because these are immediately relevant after newborn screening. Older children, adolescents, and young adults develop many chronic complications: stroke, cognitive dysfunction, priapism, leg ulcers, avascular necrosis (of the femoral head or humeral head), chronic pain, retinopathy, pulmonary hypertension, acute kidney injury, chronic kidney disease, thromboembolic events, and hepatic sequestration [50,51], cholelithiasis (gallstones) and cholecystitis as a result of excessive production and precipitation of bilirubin due to haemolysis. SCD also increases susceptibility to complications in pregnancy [52].

### 6.7. Psychosocial Impact

SCD has a significant psychosocial impact on patients and families [53]. This mainly results from the effect of pain and symptoms on their daily lives, and society’s attitudes towards them. Cultural factors are particularly important to these problems because of beliefs and practices [54]. Furthermore, the ability of people with SCD to cope with their condition varies greatly because severity, general health, and quality of life varies greatly among individuals [55,56].

## 7. Treatment and Management

SCD causes a range of acute and long-term complications, requiring a multi-disciplinary approach, involving various medical specialists. In the United Kingdom, comprehensive SCD care is coordinated by specialist haemoglobinopathy teams [57]. Such teams play a key role in education about SCD for patients and their families, as well as guiding treatment with disease-modifying therapies, access to psychology, social and welfare support. Additionally, they coordinate screening services such as Transcranial Doppler (TCD) ultrasound monitoring in children, detection of iron overload or allo-antibody formation in individuals on transfusion programmes, and referral to specialists for major organ complications with an interest in SCD.

## 8. Management of Acute Vaso-Occlusive Crises (Pain)

Pain is the commonest acute complication of SCD, and significantly impact health-related quality of life [58]. Pain management may vary from patient to patient depending on family dynamics and individual patient thresholds or access to health care, however mild to moderate painful episodes may be treated in the home without the need of attending a health facility. Self-help psychological strategies including distraction techniques such as guided imagery can be a useful evidence-based adjunct to managing pain, and patients who utilise complementary coping strategies tend to require fewer hospitalisations [53,59]. The management strategy for pain includes 4 stages: which are assessment, treatment, reassessment, and adjustment [60]. It is also important to take into consideration (1) the severity of pain and (2) the patient’s past response to different analgesics and to follow their regular protocols to alleviate the patient’s pain.

### Supportive Care

Given the fact that pain is often triggered by infection, exposure to cold or dehydration, supportive care during these episodes involves providing hydration, warmth and treating any treating the underlying infection. Simple devices such as incentive spirometry can be critical in preventing complications such as acute chest syndrome. Longer-term infection prevention varies regionally but can involve vaccination programs and penicillin prophylaxis [61].

## 9. Disease Modifying and Curative Treatments

Currently the only available disease-modifying medications for SCD are hydroxycarbamide and l-glutamine. Both are given daily to reduce the rate of acute complications, but results vary from person to person. Another effective disease modifying therapy is blood transfusion to raise the haemoglobin for improved oxygenation in severe anaemia and also to reduce the proportion of sickle haemoglobin (HbS%) may be give as a simple top-up blood transfusion or as exchange transfusion (manual or automated). The main curative therapy is stem cell transplantation while gene therapy is in the horizon in clinical Trials.

### 9.1. Hydroxycarbamide

Hydroxycarbamide has gained widely accepted use globally [62]. Although it was originally used as a cytoreductive agent by inhibiting ribonucleotide reductase, the main mechanism through which hydroxycarbamide works in SCD is through increasing total haemoglobin concentration and HbF production [63]. Hydroxycarbamide also reduces the number of leucocytes in blood, and reduces expression of surface adhesion molecules on neutrophils, red cells and vascular endothelium resulting in improved blood flow and reducing vaso-occlusion [62]. A number of trials in adults and children have shown beneficial effects of long-term hydroxycarbamide use, including reducing the severity and frequency of crisis in children with SCD [64]. The Multi-Centre Study of Hydroxycarbamide (MSH) showed that over a two-year follow-up period, adults on hydroxycarbamide had a significantly lower frequency of painful crises compared with placebo (median 2.5 versus 4.5 respectively, *p* = 0.001), as well as lower incidence of acute chest syndrome (25 versus 51, *p* = 0.001) and lower need for blood transfusion (48 patients versus 73, *p* = 0.001). A subsequent observational study following up on participants of the MSH study over a nine-year period showed a 40% reduction in mortality amongst patients on hydroxycarbamide. Patients with SCD who have increased HbF levels suffer less pain and live longer [62]. A meta-analysis in 2007 looked at effectiveness, efficacy and toxicity of hydroxycarbamide in children with SCD and they found that HbF levels increased by about 10% and also found a significant increase in haemoglobin concentration by approximately 1%; and on average there was a decrease in hospitalisation rates by 71% as well as a decrease in the frequency of pain crisis [62,65]. Indications for Hydroxycarbamide vary according to the phenotype, age and individual practice [65]. The US National Institute of Health (NIH) evidence-based guidelines and British Society of Haematology recommend offering Hydroxycarbamide to all HbSS and HbS/β0 thalassaemia genotype children from age of 1 year even though actual practice varies widely between continents.

In adults, indications for hydroxycarbamide may include [66,67,68]:Frequent painful episodes (>3 per annum) or chronic debilitating pain not controlled by usual protocols.History of stroke or a high risk for stroke or other severe vaso-occlusive events.Severe symptomatic anaemia.History of acute chest syndrome.

Patients on hydroxycarbamide undergo regular monitoring for the development of leucopenia and/or thrombocytopenia. Hydroxycarbamide can cause birth defects in animal models, hence the caution about its use during pregnancy, but hydroxycarbamide has not yet been linked to birth defects in humans. Short term research has shown only minor side effects and the benefits of using hydroxycarbamide outweigh any short-term adverse effects [62,65].

### 9.2. l-Glutamine

Glutamine is a conditionally essential amino acid, meaning that although the body normally makes sufficient amounts, at times of stress the body’s need for glutamine increases, and in such instances, it also relies on dietary glutamine to meet this demand. The U.S. Food and Drug Administration (FDA) approved use of pharmaceutical-grade l-glutamine for sickle patients aged five years or older in July 2017 [69]. Formal clinical trials showed that this purified version of glutamine significantly reduced the frequency of acute complications of SCD. Side effects appear to be minor and do not require lab monitoring [31,69].

FDA approval was based on the results of two double-blind randomized placebo-controlled trials studying the effect of l-glutamine on clinical end-points in adults and children over five years old with HbSS or *HbS/βo* thalassaemia. A phase III double-blind placebo-controlled trial randomising two hundred and thirty patients aged 5 to 58 years in a 2:1 ratio to either 0.3 g/kg oral l-glutamine twice daily, rounded to 5 g doses to a maximum of 30 g, or placebo. Concerns around the study results are due to the high dropout rate, with 97 (63.8%) participants in the intervention group and 59 (75.6%) in the placebo group completing the eleven-month study. The researchers showed a 17.9% reduction in the mean frequency of sickle crises in the l-glutamine group (3.2 versus 3.9 in the l-glutamine and placebo groups respectively (*p* = 0.0152)) and significantly fewer pain crises in the l-glutamine group (median 3.0 in the l-glutamine group and 4.0 in the placebo group, *p* = 0.005). There was significantly fewer hospitalisations in the l-glutamine group (median 2.0 in the l-glutamine group and 3.0 in the placebo group, *p* = 0.005) [69]. There were also significant reductions in the frequency of acute chest crises (8.6% on the l-glutamine group versus 23.1% in the placebo group, *p* = 0.003), and duration of hospital admissions (median 6.5 in the l-glutamine group versus 11 in the placebo group, *p* = 0.02) [69]. Potential concerns around use include the lack of long-term follow up data, financial cost compared with hydroxycarbamide, and theoretical concern around reducing treatment concordance with hydroxycarbamide therapy amongst patients seeking more naturalistic medication [70].

### 9.3. Blood Transfusion

Individuals with SCD have a baseline level of anaemia due to their chronic haemolysis. Blood transfusions are not given to correct this baseline anaemia or for acute pain episodes. Instead, transfusions are given to correct acute severe anaemia where the haemoglobin falls significantly below that individual’s baseline, and the resulting impairment in oxygen delivery to body tissues would otherwise propagate further sickling of deoxygenated Hb. Examples include red cell aplasia caused by Parvovirus B19 infection, acute splenic sequestration or hyperhaemolysis crises [71]. In an acute setting, transfusion is also used to bridge periods of severe physiologic stress like major surgery or critical illness including acute chest crises. In this setting, blood transfusion with HbS-negative blood reduces the proportion of circulating haemoglobin that is able to sickle, and hence reducing vessel occlusion and haemolysis from abnormal sickle RBCs. Long-term transfusions are instituted as a disease-modifying treatment in specific situations, such as to prevent stroke. The issues that arise as a result of long term transfusion complications include: (i) Allo-immunization, where after receiving a blood transfusion an individual develops antibodies to an antigen on the transfused red cells, which can increase the risk of having haemolytic reactions to blood that they are transfused in future; (ii) Iron overload, although treatment of iron overload is becoming more tolerable with the new oral chelators and, (iii) Risk of transfusion-transmitted infections, especially in countries where only limited screening of donated blood is available [39,72].

### 9.4. Bone Marrow Transplantation (BMT)

BMT is the only current cure for SCD and is one of the newer methods of treatments available. Results indicate an event-free survival rate of approximately 91% and a mortality rate of less than 5% [51]. BMT carries significant risks, such as the new bone marrow producing leucocytes attacking hosts tissue cells which is known as Graft-versus-host-disease (GVHD) [73]. Tissues affected include skin, liver, gastrointestinal tract and eyes, symptoms include nausea, weight loss and jaundice. The risk of developing GVHD is low when the donor and the recipient are related and matched for HLA type. When the donor and recipient aren’t related or there is a mismatch in HLA types, there is greater the likelihood of developing GVHD; strategies for careful immunosuppression after transplant can reduce the risk of GVHD [74]. Other risks from undergoing BMT include strokes, fatal infection, organ damage, and fits. Thus, BMT requires specialist centres with highly experienced teams and advanced technological resources. Due to the current levels of risk, a bone marrow transplant is only usually recommended if the symptoms and complications of SCD are severe enough to warrant the risks of BMT [75].

## 10. New and Emerging Therapies for Sickle Cell Disease

Researchers are studying several novel and existing medicines for SCD, in order to address different pathophysiological mechanisms. Candidates in the “pipeline” of clinical studies include adhesion blockers, HbS polymerization blockers, antioxidants, regulators of inflammation and activation, and promoters for nitric oxide. Developing a range of mechanistic targets may allow combination therapies to be developed, both to prevent and to treat acute sickle cell complications. For this reason, many phase II and III studies have included patients already established on hydroxycarbamide, to see if combination treatments can provide additional benefit. Examples of some of the key agents under investigation are discussed below.

### 10.1. Treatments that Reduce HbS Polymerisation

GBT440 (Voxelotor) is an oral small molecule designed to increase the oxygen affinity of HbS, shifting the oxygen dissociation curve of oxy-HbS to the left [76]. It does this by reversibly binding with the N-terminal valine of alpha (α) chain of Haemoglobin, changing its conformational structure and stabilizing the oxygenation form of the molecule. This reduces the concentration of deoxygenated HbS, which is the form of the molecule that polymerises to give the sickle phenotype. An initial phase I/II study showed Voxelotor to be well tolerated, with predictable pharmacodynamics and pharmacokinetics [76].

### 10.2. Nutritional Supplements

Omega-3 fatty acids have been purified from fish oil and tested for benefits as antioxidant, antithrombotic, and anti-inflammatory benefit. The clinical trials have used products with different purity, different proportions of types of omega-3 fatty acids, and different dosages. A trial in Georgia showed significantly decreased pain and decreased platelet activation in adults with sickle cell anemia on large daily quantities of fish oil capsules compared to adults on large daily quantities of olive oil capsules as placebo control [31,77,78]. A large trial in Sudan showed school absences were significantly decreased in children with sickle cell anemia taking fish oil compared to those taking placebo [77].

Folic acid is widely prescribed for SCD with the rationale that increased erythropoiesis causes increased risk of folate deficiency. A Cochrane Review evaluated the one double-blinded placebo-controlled clinical trial that was conducted in the 1980’s and concluded that the trial presents mixed evidence of benefit in children and no trials were found in adults [79]. Although the Cochrane reviewers recommend further investigation, they also state that no further trials of folic acid in SCD are expected [79].

EvenFlo, an herbal mixture marketed online by Healing Blends, is the only one of several herbal supplements to begin clinical trials. An open-label observational study was published online by the company in 2017 [78] and https://healingblendsglobal.com/2017/02/clinical-study-evenflo/. A randomized controlled trial was recently completed and submitted for peer reviewed publication, still pending at the time of this writing.

A traditional herbal product used to treat SCD in Nigeria, Niprisan, showed promising pre-clinical data although it is likely to have drug interactions from its significant inhibition of cytochrome CYP3A4 activity. Niprisan was awarded “Orphan Drug Status” by the FDA [32,80]. However, financial barriers halted production and Niprisan has not progressed to clinical trials [81].

### 10.3. Agents that Reduce Cell Adhesion to Activated Microvascular Endothelium: Targeted Selectin Inhibitors (Crizanlizumab, Rivipansel, Heparins and Heparin-Derived Molecules)

Selectins are transmembrane glycoproteins that are important for cell trafficking for the innate immune system, lymphocytes and platelets. Different families of selectins are expressed on endothelial cells, leucocytes and platelets. Leucocyte rolling and tethering by P and E-selectin, expressed on the surface of activated microvascular endothelium may contribute to reduced blood flow velocity and increased sickling and vaso-occlusion. Therefore, targeted P-selectin inhibitors (Crizanlizumab (SEG101, previously SELG1)), pan-selectin inhibitors (Rivipansel/GMI-1070) have undergone phase II trials. Crizanlizumab was evaluated in a double-blind, randomized, placebo-controlled phase II trial which assigned participants aged 16 to 65 years to either low-dose intravenous Crizanlizumab (2.5 mg/kg), high-dose intravenous Crizanlizumab (5.0 mg/kg), or placebo, administered over 30 min, 14 times throughout the course of one year. Results showed that a 43 percent relative risk reduction in annual acute pain episodes (1.63 vs. 2.98) occurred when comparing the high-dose Crizanlizumab and placebo treatment groups (*p* = 0.01). Not only did the therapy decrease the annual acute pain episode rate, but the high dose of Crizanlizumab also delayed the first and second acute pain episodes when compared with placebo (first episode, 4.07 months vs. 1.38 months, *p* = 0.001; second episode, 10.32 months vs. 5.09 months, *p* = 0.02; respectively [31,82].

Heparins are able to bind selectins and it has been posited that this allows heparins to reduce sickle cell adhesion to activated endothelium. A double-blind placebo-controlled randomised trial reported a statistically significant reduction in duration of painful crisis and duration of hospital admission with use of tinzaparin, a low molecular weight heparin, compared with supportive care only, amongst 253 patients with SCD admitted with acute painful crisis. Sevuparin is a novel heparin-derived compound in which the anticoagulant effect has been removed, but which retains the selectin-binding effect of heparins [83,84].

### 10.4. Agents that Improve Blood Flow through Anticoagulant Effect: Antiplatelet and Anticoagulant Agents

#### 10.4.1. Prasugrel

Prasugrel inhibits ADP-mediated platelet aggregation. Previous research suggested that activated platelets adhere to endothelium during vaso-occlusive episodes and recruit leucocytes. A phase III study of 341 children with SCD did not show a significant reduction in vaso-occlusive events per person-year in children taking Prasugrel compared with those taking placebo. It also did not show a significant reduction in diary-reported pain events [85,86].

#### 10.4.2. Apixaban

Apixaban is an oral direct Factor Xa inhibitor, which therefore prevents the activation of prothrombin to thrombin. A phase III randomised placebo-controlled trial is underway investigating the effectiveness of prophylactic dose Apixaban at reducing mean daily pain scores in adults with SCD [87].

### 10.5. Agents that Restore Depleted Nitric Oxide within the Microvasculature: Statins, l-Arginine, PDE9

Nitric oxide released from endothelium promotes vessel smooth muscle relaxation, resulting in vasodilatation and improved blood flow. It also suppresses platelet aggregation, as well as reducing expression of cell adhesion molecules on endothelium and reducing release of procoagulant factors. Intravascular haemolysis results in release of free haemoglobin into the patient’s plasma [63]. This acts as a nitric oxide scavenger. In addition, arginase released from lysed red cells breaks down arginine, which is a substrate used to make endogenous nitric oxide. Both these events result in depletion of nitric oxide levels. Two drug groups that have been investigated for their effect on improving nitric oxide reserves in SCD are statins and l-arginine. Statins inhibit Rho kinase resulting in endothelial nitric oxide synthase activation [35,88].

### 10.6. Gene Therapy

Gene therapy is in early studies as a possible cure for sickle cell anaemia. The approach is based on stem cells and gene therapy; instead of using embryonic stem cells, host stem cells are derived by manipulating and reprogramming cells from patient’s own blood cells with genetic engineering used to correct the inborn genetic error. Because the cells are provided by the patient, there is no need to find another person to serve as a donor of stem cells and there should be no risk of GVHD. The aim is to transform a patient’s blood cells into pluripotent stem cells and replace the defective portion of the gene. These cells will then be coaxed into becoming hematopoietic cells which can specifically regenerate the entire range of red blood cells. At the time of this writing, a handful of people have apparently been cured of SCD in three gene therapy clinical studies with different lentiviral vectors [89].

A number of new sickle cell therapeutic options are on the horizon; the promise of combination therapy is no longer a far-fetched aspiration. This calls for an urgent debate with regards to the correct combinations, the right patient phenotype and access for the majority of patients. It is therefore timely to commission such a review on newborn sickle cell screening not just for European countries which of course face the migration challenge, but also Africa and India [90].

## Figures and Tables

**Figure 1 IJNS-05-00020-f001:**
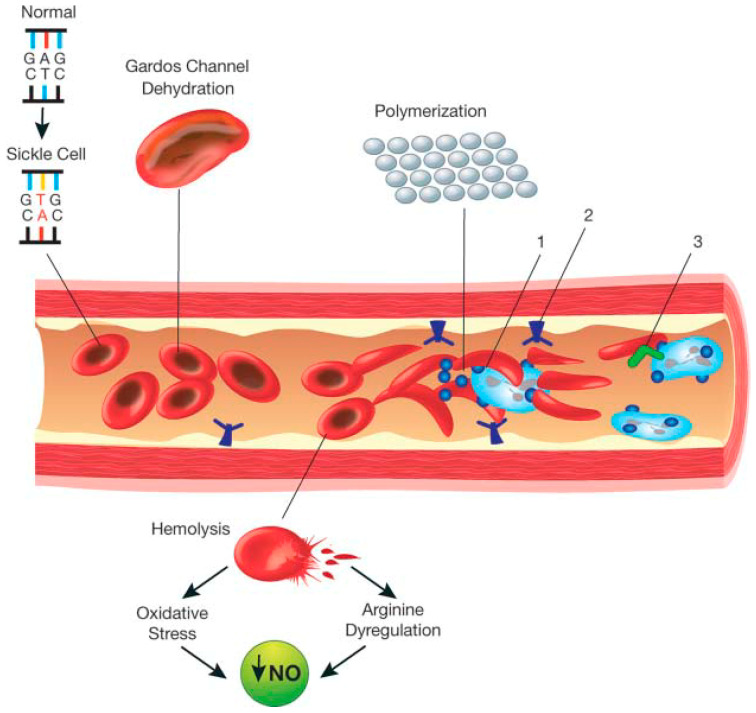
Schematic representation of the pathophysiology (in part) of sickle cell anemia. A single gene mutation (GAG→GTG and CTC→CAC) results in a defective haemoglobin that when exposed to de-oxygenation (depicted in the right half of the diagram) polymerizes (upper right of the diagram), resulting in the formation of sickle cells. Vaso-occlusion can then occur. The disorder is also characterized by abnormal adhesive properties of sickle cells; peripheral blood mononuclear cells (depicted in light blue; shown as the large cells under the sickle cells) and platelets (depicted in dark blue; shown as the dark circular shapes on the mononuclear cells) adhere to the sickled erythrocytes. This aggregate is labelled 1. The mononuclear cells have receptors (e.g., CD44 (labeled 3 and depicted in dark green on the cell surface)) that bind to ligands, such as P-selectin (labeled 2 and shown on the endothelial surface), that are unregulated. The sickle erythrocytes can also adhere directly to the endothelium. Abnormal movement or rolling and slowing of cells in the blood also can occur. These changes result in endothelial damage. The sickled red cells also become dehydrated as a result of abnormalities in the Gardos channel. Hemolysis contributes to oxidative stress and dysregulation of arginine metabolism, both of which lead to a decrease in nitric oxide (NO) that, in turn, contributes to the vasculopathy that characterizes SCD.

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
