# Peer review of "Sickle Cell Disease—Genetics, Pathophysiology, Clinical Presentation and Treatment"

_2409-515X, 2019, doi:10.3390/ijns5020020_

Reviewer 1 Report

The authors of the paper did a complete and significant overview of the clinical feature, the epidemiology and an update on present and future  experimental treatments for Sickle Cell Disease.

 I have some minor comments:

 Line 62: Please mention other publications, for example:

1.      Significant prevalence of sickle cell disease in Southwest Germany: results from a birth cohort study indicate the necessity for newborn screening. Kunz JB, Awad S, Happich M, Muckenthaler L, Lindner M, Gramer G, Okun JG, Hoffmann GF, Bruckner T, Muckenthaler MU, Kulozik AE. Ann Hematol. 2016 Feb;95(3):397-402. doi: 10.1007/s00277-015-2573-y. Epub 2015 Dec 12.

2.      The Prevalence of Sickle Cell Disease and Its Implication for Newborn Screening in Germany (Hamburg Metropolitan Area). Grosse R, Lukacs Z, Cobos PN, Oyen F, Ehmen C, Muntau B, Timmann C, Noack B. Pediatr Blood Cancer. 2016 Jan;63(1):168-70. doi: 10.1002/pbc.25706. Epub 2015 Aug 14.

3.      Results of a multicenter universal newborn screening program for sickle cell disease in Italy: A call to action. Colombatti RMartella MCattaneo LViola GCappellari ABergamo CAzzena SSchiavon SBaraldi EDalla Barba BTrafojer UCorti P, Uggeri MTagliabue PEZorloni CBracchi MBiondi ABasso GMasera NSainati L. Pediatr Blood Cancer. 2019 Feb 5:e27657. doi: 10.1002/pbc.27657.

Lines 119-133: Are the data concerning the sickle haplotypes and the genetic variations of BCL11A, HBS1L-MYB etc involved on the HbF levels  useful in the clinical management of the disease, for example for dosage of hydroxycarbamide? Do the authors have experience in this issue?

Line 192-194:   Cross check position editing.

Line 237: ”Strouse et al 2007” is absent in the list of references.

“Strouse et al (2007) meta-analysis looked at effectiveness, efficacy and toxicity of Hydroxycarbamide in children with SCD and they found that HbS levels increased by about 10% and also found a significant increase in hemoglobin concentration by approximately 1%”.

My question is : “increased or decreased” HbS levels?

Line 252-257:  Please mention other publications, for example:

1.      Side effects of hydroxyurea in patients with Thalassemia major and thalassemia intermedia and sickle cell anemia. Ghasemi AKeikhaei BGhodsi R. Iran J Ped Hematol Oncol. 2014;4(3):114-7. Epub 2014 Jul 20.

Line 410, 463, 497 and 503: The references are incomplete.

Author Response

Line 62: Please mention other publications, for example:

1.      Significant prevalence of sickle cell disease in Southwest Germany: results from a birth cohort study indicate the necessity for newborn screening. Kunz JB, Awad S, Happich M, Muckenthaler L, Lindner M, Gramer G, Okun JG, Hoffmann GF, Bruckner T, Muckenthaler MU, Kulozik AE. Ann Hematol. 2016 Feb;95(3):397-402. doi: 10.1007/s00277-015-2573-y. Epub 2015 Dec 12.

2.      The Prevalence of Sickle Cell Disease and Its Implication for Newborn Screening in Germany (Hamburg Metropolitan Area). Grosse R, Lukacs Z, Cobos PN, Oyen F, Ehmen C, Muntau B, Timmann C, Noack B. Pediatr Blood Cancer. 2016 Jan;63(1):168-70. doi: 10.1002/pbc.25706. Epub 2015 Aug 14.

3.      Results of a multicenter universal newborn screening program for sickle cell disease in Italy: A call to action. Colombatti R, Martella M, Cattaneo L, Viola G, Cappellari A, Bergamo C, Azzena S, Schiavon S, Baraldi E, Dalla Barba B, Trafojer U, Corti P, Uggeri M, Tagliabue PE, Zorloni C, Bracchi M, Biondi A, Basso G, Masera N, Sainati L. Pediatr Blood Cancer. 2019 Feb 5:e27657. doi: 10.1002/pbc.27657.

Our response

We have now updated all the relevant publications including the ones above suggested by the reviewer. This has in fact enriched our manuscript, they are now included in the full report.

Lines 119-133: Are the data concerning the sickle haplotypes and the genetic variations of BCL11A, HBS1L-MYB etc involved on the HbF levels  useful in the clinical management of the disease, for example for dosage of hydroxycarbamide? Do the authors have experience in this issue?

Our response. Due to the fact that our paper is an overview we are limited in the data relating to hydroxyurea therapy. We acknowledge that the response to hydroxyurea appear to correlated mainly to the level of pre-existing foetal haemoglobin level and this is discussed in relation to the sickle haplotype by Piel et al, NEJM review and it is referenced in the write up.

Line 192-194:   Cross check position editing.

Our response – we have made substantial review of the manuscript and addressed this issue fully

Line 237: ”Strouse et al 2007” is absent in the list of references.

“Strouse et al (2007) meta-analysis looked at effectiveness, efficacy and toxicity of Hydroxycarbamide in children with SCD and they found that HbS levels increased by about 10% and also found a significant increase in hemoglobin concentration by approximately 1%”.

Our response, this has been rectified accordingly.

My question is : “increased or decreased” HbS levels?

Line 252-257:  Please mention other publications, for example:

1.      Side effects of hydroxyurea in patients with Thalassemia major and thalassemia intermedia and sickle cell anemia. Ghasemi A, Keikhaei B, Ghodsi R. Iran J Ped Hematol Oncol. 2014;4(3):114-7. Epub 2014 Jul 20.

Line 410, 463, 497 and 503: The references are incomplete.

The incomplete references has been fully addressed, this was due to a software glitch which is now fixed.

Reviewer 2 Report

Genetics, Pathophysiology, Clinical Presentation and Treatment

By Baba Inusa et al.

This is timely, important and interesting review. Unfortunately numerous typos and stylistic errors make manuscript difficult to read. The format and size of the letters are different through the text. Also several chapters are very sketchy and not well developed that significantly reduces the potential impact of the review. The manuscript should be significantly revised before acceptance for publication.

Major comments:

1.     Numerous terms in the text didn’t define before use. For example, Line 184 what is sRBC? Line 229: authors switched from the term ”hydroxycarbamide” to “ hydroxyurea” several times  in the chapter 8.2.

2.     Chapters 6.2 (anemia) and 6.6 (other organs complications) are very sketchy and should be extended.  

3.     The most interesting and novel chapter 9 should be re-write (correct typos; re-numerate; remove repeats etc.)

4.     Conclusion should be improved in the context of current SCD initiatives and prospects.

Minor comments:

1.     There are numerous typos in the text. Please read and correct them.

Author Response

Reviewer 2

This is timely, important and interesting review. Unfortunately numerous typos and stylistic errors make manuscript difficult to read. The format and size of the letters are different through the text. Also several chapters are very sketchy and not well developed that significantly reduces the potential impact of the review. The manuscript should be significantly revised before acceptance for publication.

Major comments:

1.     Numerous terms in the text didn’t define before use. For example, Line 184 what is sRBC? Line 229: authors switched from the term ”hydroxycarbamide” to “hydroxyurea” several times in the chapter 8.2.

A wholesome review of the manuscript has been done to rectify all the typos. We have changed been consistent in using hydroxycarbamide

2.     Chapters 6.2 (anemia) and 6.6 (other organs complications) are very sketchy and should be extended

-We acknowledge the issue, providing more details on key subjects.

3.     the most interesting and novel chapter 9 should be re-write (correct typos; re-numerate; remove repeats etc.)

- This chapter on novel therapies has been re-written.

4.     Conclusion should be improved in the context of current SCD initiatives and prospects.

Minor comments:

1.     There are numerous typos in the text. Please read and correct them.

- Our response, we have fixed the errors and re-written most sections. We agree that this has improved the whole manuscript, it reads a lot better.

Round  2

Reviewer 2 Report

Authors replied all comments and extensively corrected text that significantly improved manuscript. I recommend to accept the manuscript for publication.